# Evaluation of Strong Cation Ion-Exchange Resin Cost Efficiency in Manufacturing Applications—A Case Study

**DOI:** 10.3390/polym14122391

**Published:** 2022-06-13

**Authors:** Maciej Jerzy Kobielski, Wojciech Skarka, Maciej Mazur, Damian Kądzielawa

**Affiliations:** 1Sanhua-Aweco, Turyńska 80, 43-100 Tychy, Poland; maciej.mazur@sanhua-aweco.com (M.M.); damian.kadzielawa@sanhua-aweco.com (D.K.); 2Department of Fundamentals of Machinery Design, The Faculty of Mechanical Engineering, Silesian University of Technology, Stanislawa Konarskiego 18A, 44-100 Gliwice, Poland; wojciech.skarka@polsl.pl

**Keywords:** ion-exchange resin, water softening, cost efficiency, cost-based design optimization, energy efficiency, cost-efficiency, effective ionic capacity, material applicability evaluation, behavioral difference

## Abstract

The effective ionic capacities of strong cation ion-exchange resins were investigated and compared using conditions similar to those found in white goods, in order to establish behavioral differences between commercial products and evaluate their capacity in a broader business context. Nine different products of equivalent TDS (Technical Data Sheet) capacity were observed to examine their differences in approximately real-life conditions. For a broader context of applicability analysis, besides the absolute ionic operating capacity, the following additional factors were included in the evaluation: the standard deviation in the resins’ performances and their relative prices. A complete method for material applicability evaluation was hereby proposed and shown to offer cost factor benefits of up to 21.1% within the range of products examined, in comparison to a cost-only evaluation for equivalent materials.

## 1. Introduction

A large proportion of market-available dishwashers are designed with a water softening device, and not without good reasons. The high calcium and magnesium ion content in process water can lead to the damage of internal hydraulic components by the formation of limescale [1]. It can lead to a drop in the energy efficiency of the dishwashing process via limescale insulation on heating surfaces [2]. Additionally, limescale creation inside a dishwasher can pose a hygiene risk, should the roughness of the surfaces be increased by these deposits [3,4,5]. The cleaning efficiency will drop if hard water is used for dishwashing, as dirt removal has been observed to deteriorate in the presence of Ca^2+^ ions [6]. Whereas it could be indicated that small amounts of Ca^2+^ ions can positively affect the cleaning efficiency by easing the formation of micelles by detergent’s anionic surfactants which are critical to the chemical process of soil removal [7], the anionic particles of the surfactant carry a negative charge on the hydrophilic side and can be deactivated by the presence of Ca^2+^ ions [8] Finally, in the presence of hard water ions, the foaming process of the detergents will also be affected [6].

The benefits of effective softening in dishwashing, therefore, have a direct impact on its carbon footprint: according to the ASTM D3556 (Standard Guide for Deposition on Glassware During Mechanical Dishwashing ) dishwasher testing method, it was found that the use of softened water can allow for detergent savings of up to 70%, significant improvement with respect to dirt elimination at lower temperatures, and an overall drop in energy consumption by the dishwasher in the long term [9].

There is a lot of research available on deionization methods, including ion-exchange resins [10,11], zeolites, super-macroporous cryogels [12], and nanofiltration membrane method [13]. Recently, a lot of research has been carried out on capacitive deionization methods [14,15,16] using flow-through capacitors [17] or reverse osmosis. According to the data, membrane methods are the most popular for industrial applications [18]; however, the cost of these systems remains high in comparison to ion-exchange resins [19].

As the use of ion-exchange resins is the industry standard for dishwashing systems, the market offers a variety of off-the-shelf strong acid cation-exchange resins for Ca^2+^ and Mg^2+^ ion elimination, regenerated by NaCl solution.

The materials’ operational capacity would usually be defined by the equivalent unit—‘eq’—but this method has severe drawbacks in terms of the evaluation of operational efficiency [20].

For the commercial success of a design in the household appliance industry, it is necessary to ensure the best cost-to-performance ratio of products, considering the behavioral differences between nominally comparable configurations. These differences create a need for a robust comparison method, taking into consideration both the chemical behavior of the material under the given conditions, as well as the cost-related aspect of the product application.

The purpose of this paper was to present a new method for strong acid cation-exchange resin comparison, which allows raw material costs regarding production-level efficiency to be compared.

The method is intended for use in further research on household appliance ion-exchange production cost optimization, in parallel with the authors’ model-based systems engineering (MBSE) methodology [21,22,23], effectively used for evaluation, optimization, and improvements in robustness and efficiency.

## 2. Materials and Methods

### 2.1. Material Products and Samples

A comparison method was devised with the goal of effectively evening out the differences between the product samples and achieving an objective comparison of the efficiency when applied in conditions similar to those found in normal dishwasher operation.

The hydrodynamically normalized resin samples were operated in an ultra-hard water environment and were set up to dynamically exchange ions in an automated process with precisely prepared comparable conditions.

An overview of the process flow is shown in Figure 1.

#### 2.1.1. Resin Products

The strong acid cation-exchange resins used for this case study were provided as free-of-charge test samples by market-leading producers. All the suppliers’ names and the ion-exchange resin products’ brand names were omitted due to commercial considerations.

A total of nine commercial-grade resin samples from seven producers were used in the experiments: all the samples were strong acid cation-exchange resins of 1.9–2.0 eq/L capacity, regenerable by NaCl solution, with a comparable bead distribution of 0.32–1 mm for the mitigation of the impact of the pressure gradient on the chemical ion capacity results. All the resins were of gel type, ST-DVB (Styrene-divinylbenzene) matrix based, with functional groups of sulfonic acid—shelf-available as industrial water softening ion exchange resins. There was no difference in listed chemical composition betwixt the compared specimens. The maximum water retention in the Na^+^ state of the tested material samples is shown in Figure 2.

No resins besides the listed types were in the scope of this study due to specific customer requirements in the dishwasher industry.

#### 2.1.2. Sample Geometry

The material resin samples were placed in a wide, cylindrically shaped column designed specifically for the tests (Figure 3). The column geometry was designed so that the internal volume was 515 mL, housing a volume of 500 mL of beaded resin and 3% free space for the possible movement of the resin beads under flow conditions.

Each resin column consisted of an 84 mm diameter PMMA (Polymethyl Methacrylate) tube of 140 mm length, fitted with two injection-molded polypropylene sieves having 0.3 mm slit width and radial orientation of the gaps. The resin column was fitted with two identical, radial flow constraining caps, designed specifically for this application and FFF (Fused Filament Fabrication) printed using polypropylene. The samples were later radially compressed to achieve external tightness with the use of electrical insulation tape.

The geometry was devised so that the column would operate in a vertical position to allow for flow in the direction opposite to gravity, and to simulate real-life application conditions.

#### 2.1.3. Sample Normalization

To eliminate the impact of the weight of residual water in the resin bead structure and the inter-bead annular space for each material sample, the resins were weighed in a laboratory cylinder completely submerged in water. The resin sample volumetric density was evaluated under complete water saturation conditions to allow for comparable space use in the apparatus. For each resin product, the maximum water saturation volumetric density was measured and compared to the weight on delivery, to be used as an indication for weight-based dosing of each sample, to ensure 500 mL of beaded resin volume under sample flow (submersion) conditions.

### 2.2. Reagents

#### 2.2.1. Hard Water Preparation

Pure magnesium chloride hexahydrate and pure calcium chloride hexahydrate, used for artificially preparing standardized ultra-hard water for testing purposes, were obtained from Sigma-Aldrich (St. Louis, MO, USA). The initial non-hardened water was obtained from an in-house industrial water deionization installation.

The water for testing was prepared in a 0.7 m^3^ container by use of the abovementioned chloride salts after a prior control of the initial calcium and magnesium hardness by the EDTA (Ethylenediaminetetraacetic Acid) method and subsequent addition of the chloride salts in calculated amounts based on the desired molar contents according to the German hardness scale, using the MgCl_2_ and CaCl_2_ in the following proportions (Equation (1)):mass (MgCl_2_) [g] = 2.244 × mass (CaCl_2_) [g](1)

#### 2.2.2. Water Hardness Control—Edetate Method

The total permanent hardness of the water was controlled according to the German hardness scale; it was tested using the EDTA titrimetric method. Other reagents—a 25% solution of NH_4_OH, Eriochrome Black T, pure NaCl, and 0.005 M EDTA solution—were all purchased from Sigma-Aldrich (St. Louis, MO, USA).

The method was used in accordance with Standard Methods for the Examination of Water and Wastewater [24].

#### 2.2.3. Residual Brine Control by an Argentometric Method

The ion-exchange resins’ evaluation by analysis of their residual water hardness was supplemented by control of the brine parameters after regeneration. Reagents for the method—consisting of silver nitrate solution as titrant and potassium chromate as indicator solution—were purchased from Sigma-Aldrich (St. Louis, MO, USA). Deionized water was obtained from an in-house industrial deionization installation (Sanhua-Aweco, Tychy, Poland).

The method was used in accordance with Standard Methods for the Examination of Water and Wastewater [24].

#### 2.2.4. Regeneration

The regeneration process for the ion-exchange resin was based on household-appliance grade fine NaCl salt, Ludwik (INCO Group, Warsaw, Poland) brand, purchased from Lyreco (Marly, France).

### 2.3. Equipment

#### 2.3.1. Testing Equipment

The experiments were conducted using a proprietary dedicated resin control station (Figure 4, Sanhua-Aweco GmbH, Neukirch, Germany based on LabView software and National Instrument DAQ (Data Acquisition) modules. The system consisted of a 700 L hard water tank, a submersion pump for the hard water tank, a set of flowmeters to control the water flow, a dedicated external 20 L container for the saturated regeneration NaCl brine, a submersion pump for the brine container, a gravity overflow buffer container for volumetric regulation of the regeneration brine (which was kept at a fixed amount), a system-controlled switch valve to alternate between feeds in a timely manner, a treated water mixing buffer for ensuring a consistent ion concentration in treated water samples, and a CNC feed arm from the mixing buffer that allowed for alternating the sampling position between 32 sampling beakers (200 mL). All the electrical components used dedicated power supplies, controlled by the DAQ module and the software. The purpose of the setup was to ensure very high repeatability for the resin testing, which could also be realized by manual control methods, according to according to the description in Section 2.4.2.

#### 2.3.2. Titration Equipment

The water control, which was carried out according to standard methods [24], was performed using the following titration equipment (Figure 5):Two BRAND^®^ Titrette^®^ 25 mL digital bottle-top burettes; (BRAND GMBH + CO KG Wertheim, Germany)BRAND^®^ Dispensette^®^ S, fixed volume, 10 mL; (BRAND GMBH + CO KG Wertheim, Germany)BRAND^®^ fixed volume micropipette; (BRAND GMBH + CO KG Wertheim, Germany)Laboratory magnetic stirrer. (Sanhua-Aweco GmbH, Neukirch, Germany)

### 2.4. Methods

#### 2.4.1. Water and Brine Preparation

##### Hard Water Preparation

The water tank was filled with deionized water, subject to hardness evaluation according to the edetate method. The water hardness was defined as Ca and Mg ion content in a mass proportion of 1:2.244, set by the addition of calculated amounts of magnesium chloride hexahydrate and calcium chloride hexahydrate and thorough mixing. After the initial addition of substances, a second hardness evaluation was carried out—should the tolerance of 68–70 dH not be met, control loops were used to add the requisite salts to achieve hardness tolerance.

##### Brine Preparation

The 20 L PP (Polypropylene) container was filled with water at ambient temperature, after which 3 kg of dishwasher-grade salt was added. The contents were mechanically mixed until the brine density—controlled by a glass hydrometer—showed complete saturation, with unsolved NaCl residue on the bottom of the container. The salt content was calculated based on the hydrometer density test result.

#### 2.4.2. Ion-Exchange Testing Cycle

The operational efficiency of the ion-exchange resins was evaluated based on a cycle derived from a commercial specification set for this type of material from a market-leading household appliance OEM (Original Equipment Manufacturer ) producer.

The cycle, shown in Figure 6, was initiated by a fixed amount of brine to allow for comparable cycle repetition. The brine volume was mechanically fixed at 240 mL, gravitationally filling the resin container from the bottom, and allowing for a 600 s rest time for the resin regeneration to take place. After 600 s, the brine was flushed with the previously prepared hard water, in four cycles of 1 L, with a fixed flow of 2.5 L/min. The cycles were spaced by 30 s pauses. The total amount of 4240 mL of the regeneration brine and flushing hard water was contained by a mixing buffer zone, from which a 100 mL sample was automatically drawn to the first glass beaker.

After regeneration, four softening cycles took place. All the softening cycles consisted of 2 L of 70 dH water, flushing the resin with a 2.5 L/min fixed flow. Beginning with a 70 s pause, consecutive softening cycles followed. The water from all the softening cycles was contained in a mixed buffer zone, from which 100 mL samples were taken in the following order: 100 mL was drawn from the first cycle to the second beaker, the second and third cycles were mixed for an averaged 100 mL third beaker sample, whereas the fourth cycle was sampled (100 mL) to the fourth beaker in the row.

After the fourth softening sampling, the cycle would restart in the described manner, to gather a total of 32 beaker samples from a run of eight cycles.

Each resin was subject to at least two runs, totaling a minimum of 16 regeneration-softening cycles.

#### 2.4.3. Comparative Method

A new comparative method was devised for the applicability study of the strong acid cation-exchange resin products, looking at nominally chemically equivalent samples from the perspective of economic justification.

The residual water hardness—measured by the EDTA titrimetric method, along with the known initial water ionic content—allowed for the calculation of the molar operating capacity of each tested product for each softening cycle, the capacity being measured per fixed testing volume under conditions set up for compatible comparison.

The results were averaged and divided by the sample beaded resin volume in liters, allowing for the calculation of an average molar operating capacity per 1 L of beaded submerged resin for each of the ion-exchange resin products.

The use of mass per submerged beaded resin volume data from the sample normalization procedure allowed us to recalculate the average molar operating capacity per liter of beaded resin to achieve the average molar operating capacity per kg of delivery-condition material.

In our final approach, confidential commercial price quotations for each material—including logistical costs—allowed for cost data to be calculated on a per kilogram basis. The per-kilogram price was divided by the per-kilogram delivery condition averaged molar capacity, to obtain the cost of the deionization efficiency for each material sample.

The values obtained were subject to an additional engineering-derived correction method: the standard deviation of the testing results for the volumetric capacity for each resin product was calculated as a percentage of the result, tripled, and added to the previously obtained cost according to Six Sigma methodology as the cost of a design safety buffer for ensuring the expected efficiency in 100% of products.

## 3. Results

Nine ion-exchange materials were subject to comparative testing. The results obtained, which were based on manually performed chemical testing of the beaker-stored samples, consisted of nine datasets from at least 16 runs of softening–regeneration cycles.

The ionic capacity results of the nominally equivalent resins were different under the applied conditions. A comparison of the nominally calculated molar operating capacity is shown in Table 1.

The per kg capacity comparison is shown in Figure 7.

The results of the molar capacity per kilogram are indicative of the best performing material from a chemical perspective. However, industrial cost and engineering approach requires the prices of the materials to be compared. Due to the confidential nature of commercial conditions, the prices are not displayed. The indicated cost shown was calculated based on the ratio of the actual prices, with the highest value of 1 attributed to the most expensive material and logistic costs being included, giving a comparative price for 1000 metric tons per year under DAP (Delivered at Place) Incoterms conditions for Poland.

A comparison of the materials’ relative costs is shown in Table 2 and Figure 8.

Based on the averaged tested operational capacity (Table 1 and Figure 7) and the relative price of each resin specimen (Table 2 and Figure 8), the relative cost of 1 mol operational capacity was calculated by division of each relative price by respective operational capacity converted to full molar units (Table 3 and Figure 9). Following the method described, the goal was to assess the engineering applicability cost. The results, based on the relative resin price shown in Figure 8, including the resin product behavioral correction of three standard deviations according to the Six Sigma methodology (shown in Figure 10), are shown in Table 4 and Figure 11.

## 4. Discussion

In the household appliance sector, when designing a water softening device, it is standard practice to evaluate the specific ionic capacity requirement according to the equivalent capacity value and adapt the applied container volume according to the type of resin chosen, with the material choice being dictated by the per-mass-unit cost of the quoted raw material. The capacity values provided in technical data sheets by the material manufacturers rarely correspond to capacity values achieved in normal operation, as the theoretical capacity values are parameters related to the amount of functional groups added in the resin’s production process, not to the characteristic of the material itself. The presented approach simultaneously takes into consideration the real-life application resin efficiency, the business practice for design safety, and the material pricing, allowing the optimum cost for a material’s use to be chosen.

The data from the applied resin testing showed up to a 31.4% efficiency difference in ionic retention in the applied conditions between the nine resin products. The test conditions using water with ultra-high hardness allowed us to effectively compare the chemical efficiency of the resin samples, while the preparation process allowed us to mitigate the impact of specific density and humidity differences in the as-delivered samples. At the same time, while these parameters were eliminated from impacting the hydrodynamical conditions of the comparison testing, their impact was preserved at the stage of sample cost evaluation, as well as in the complete comparison of the efficiency cost for this application.

A comparison of the prices for the nine resin products, assuming an equivalent efficiency, would result in a material choice of Resin C firstly, with the lowest relative cost of 0.7213; secondly, Resin F with a cost of 0.7426; and thirdly, Resin E—if required—with a relative price of 0.7439. A choice made in this way would have been suboptimal in light of the efficiency test results.

A comparison of the measured ionic retention efficiency and relative price, shown in Figure 10, allowed us to draw our first conclusions on the real costs for the samples and eliminate the F resin choice, placing Resin A in first place with a per-capacity relative cost of 2.9794 mol^−1^, Resin E in second with a per-capacity relative cost of 2.9803 mol^−1^, and Resin C in third with a per-capacity relative cost of 3.1637 mol^−1^.

A complete approach, including a behavioral deviation correction, allowed us to take into consideration the difference in capacity consistency regarding resin choices, changing the preference yet again. The following optimum choice order was obtained: Resin E was the optimum choice to be used in the specified application with a cost of 3.1105 mol^−1^, Resin A was the second-best with a cost of 3.1216 mol^−1^, and Resin D was third with a cost of 3.2799 mol^−1^.

Further corrections could be applied, taking into consideration the supplied materials’ statistical process control indicators; however, this kind of study would require long-term quality data collection on recurrent standardized deliveries of selected resins, which would not be justified from the perspective of material choice and the benefit to the product’s application.

## 5. Conclusions

The results of this study clearly indicated the effectiveness of the approach. The nominally preferable Resin C was measured and proven to be the fourth most economical solution, with over 21.1% higher price with respect to its effectiveness compared to Resin E, which was 3% more costly using a price per kilogram approach. This result can translate to a 21.1% material cost saving for the products, respective to cost comparison for comparable materials according to the equivalent ionic capacity indicator, ‘eq’. The comparison method in its complete version—including the correction for behavioral stability by including a standard deviation factor—provided an additional 0.35% potential product cost improvement, relative to the molar-capacity-only method, insofar as the applied target complied with the defined functional requirements.

In theory, this approach could be applied to any functional material should the material’s function be quantifiable in the desired application.

## Figures and Tables

**Figure 1 polymers-14-02391-f001:**
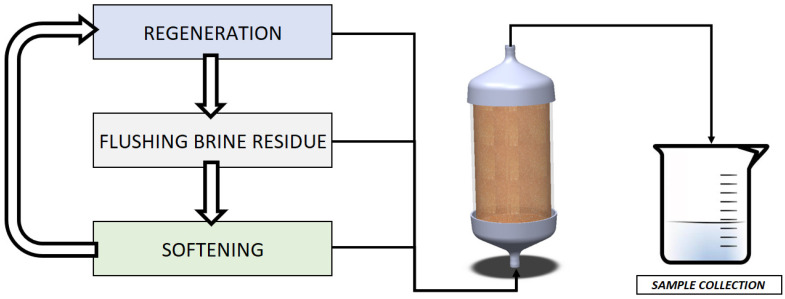
Test schematics with CAD representation of encapsulated resin.

**Figure 2 polymers-14-02391-f002:**
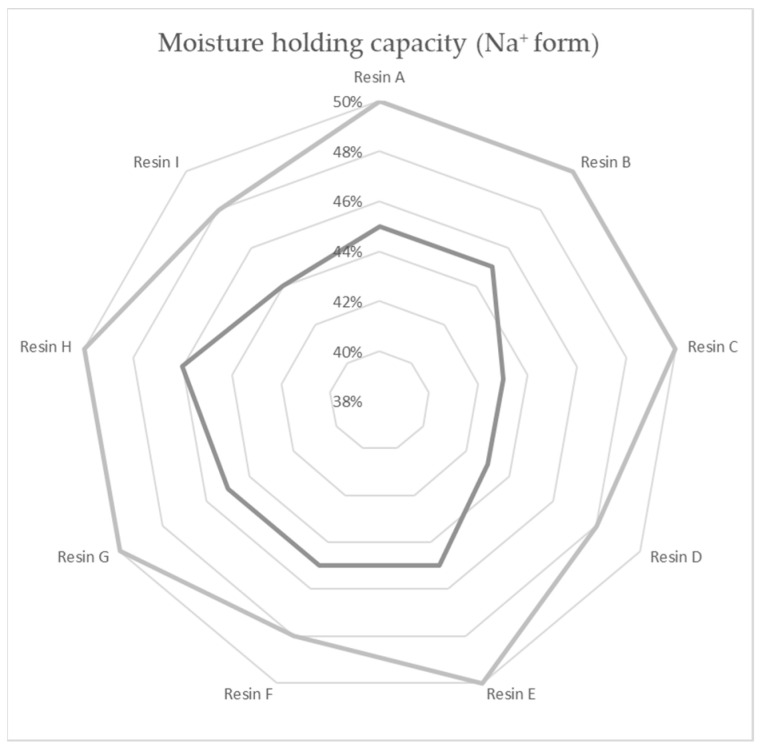
Moisture holding capacity of resin specimens according to the technical data sheets.

**Figure 3 polymers-14-02391-f003:**
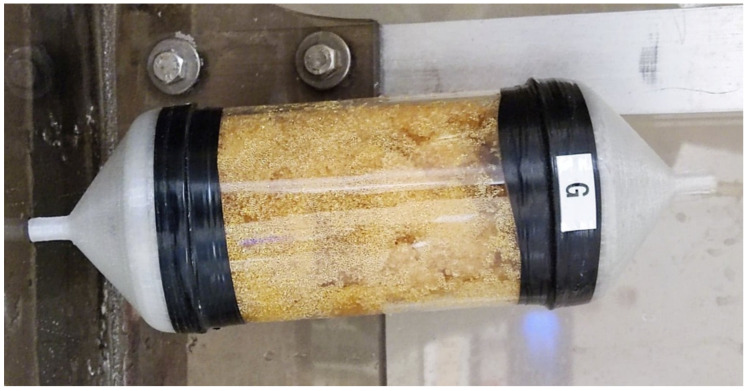
Test Sample G—representative picture.

**Figure 4 polymers-14-02391-f004:**
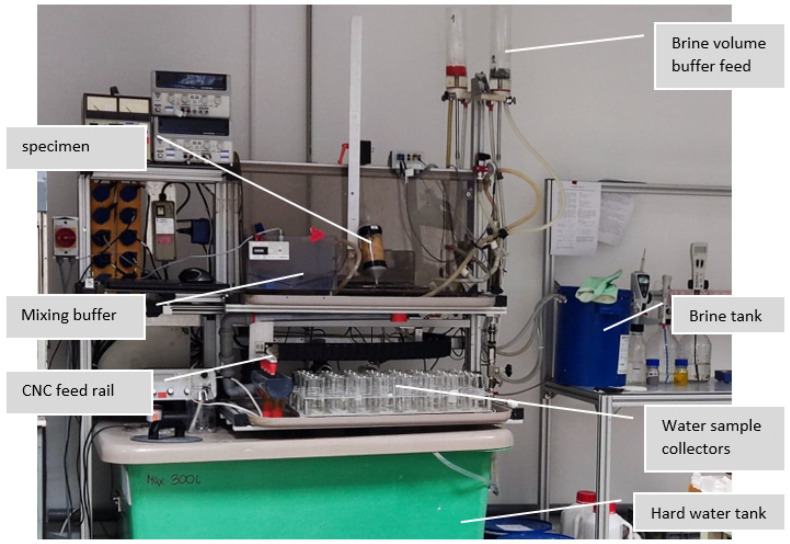
Picture of the described Sanhua Aweco proprietary resin testing equipment.

**Figure 5 polymers-14-02391-f005:**
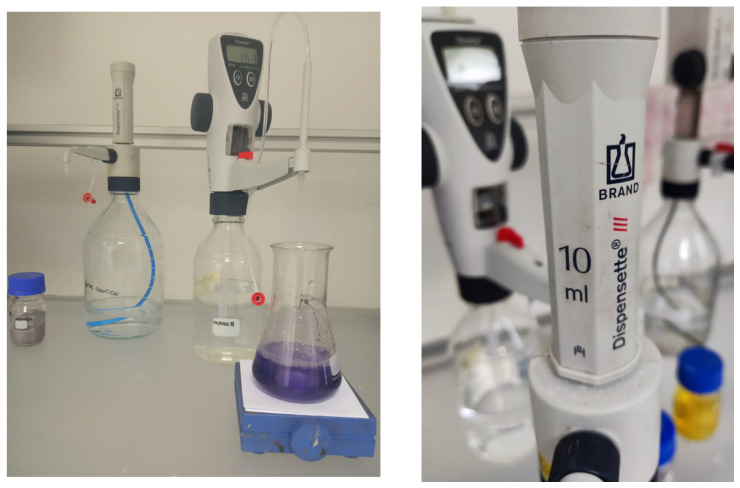
Pictures of titration equipment, EDTA method setup.

**Figure 6 polymers-14-02391-f006:**
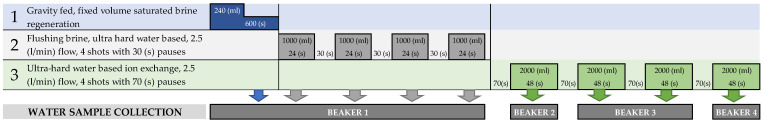
Resin test product’s timing, solution volumes, and operation logic used in the study.

**Figure 7 polymers-14-02391-f007:**
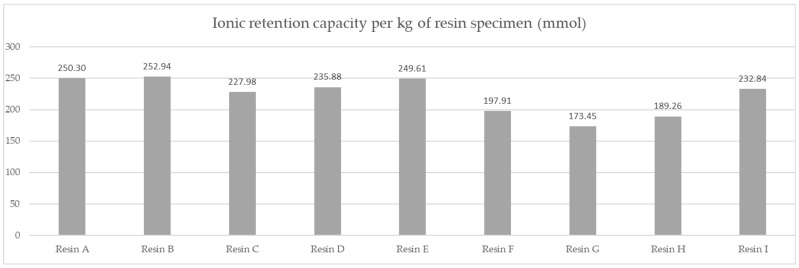
Comparison of average molar capacity in given test conditions of the resin products.

**Figure 8 polymers-14-02391-f008:**
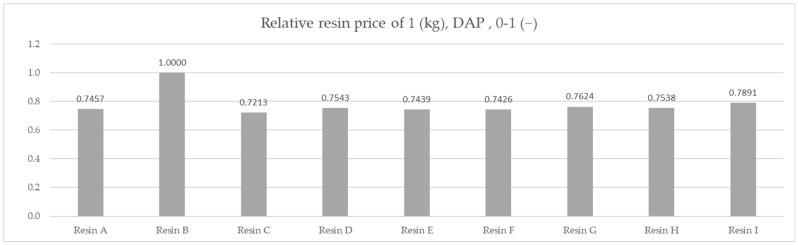
Comparison of the relative cost of 1 kg for resin products under DAP Incoterms.

**Figure 9 polymers-14-02391-f009:**
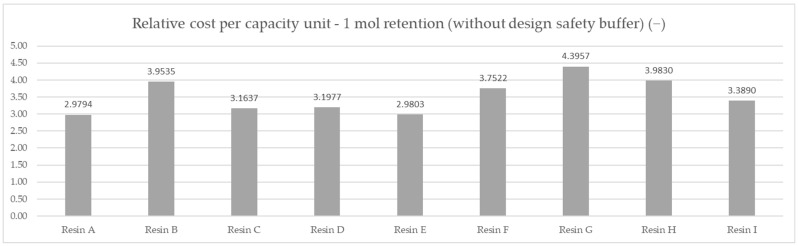
Comparison of the relative cost of calculated average mmol capacity unit—DAP price for 1 mol capacity for the set testing conditions.

**Figure 10 polymers-14-02391-f010:**
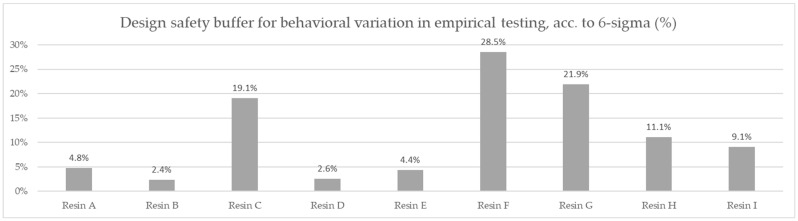
Comparison of the test-derived safety buffer required to ensure a repeatable efficiency for the tests, according to the Six Sigma methodology.

**Figure 11 polymers-14-02391-f011:**
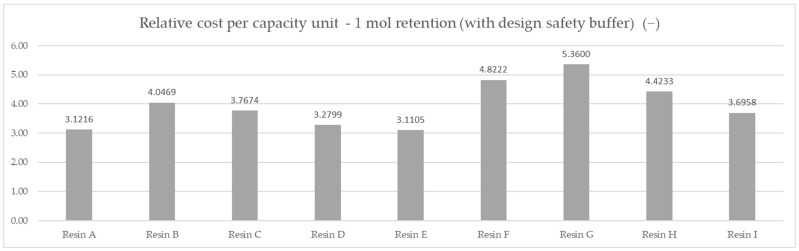
Comparison of the relative cost of the calculated average mmol capacity unit, corrected for behavioral differences—DAP price of 1 mol capacity under the applied set testing conditions, three standard deviations’ correction according to Six Sigma methodology.

**Table 1 polymers-14-02391-t001:** Comparison of calculated averaged molar capacities of resin products and test result standard deviation per sample in %.

Resin Product	Molar Capacity per Liter (mmol)	Molar Capacity per kg (mmol)	Test Std (%)
Resin A	213.39	250.30	1.591
Resin B	208.17	252.94	0.787
Resin C	201.90	227.98	6.361
Resin D	217.48	235.88	0.857
Resin E	203.68	249.61	1.457
Resin F	182.47	197.91	9.506
Resin G	161.31	173.45	7.313
Resin H	190.77	189.26	3.685
Resin I	214.68	232.84	3.017

**Table 2 polymers-14-02391-t002:** The relative cost of 1 kg for resin products under DAP Incoterms.

Resin Product	Relative Cost of 1 kg of Material
Resin A	0.7457
Resin B	1
Resin C	0.7213
Resin D	0.7543
Resin E	0.7439
Resin F	0.7426
Resin G	0.7424
Resin H	0.7538
Resin I	0.7891

**Table 3 polymers-14-02391-t003:** The relative cost of 1 mol capacity for resin products.

Resin Product	Relative Cost of 1 mol Capacity
Resin A	2.9794
Resin B	3.9535
Resin C	3.1637
Resin D	3.1977
Resin E	2.9803
Resin F	3.7522
Resin G	4.3957
Resin H	3.9830
Resin I	3.3890

**Table 4 polymers-14-02391-t004:** The relative cost of 1 mol capacity for resin products corrected for a Six Sigma safety design buffer based on test standard deviation for applied testing.

Resin Product	Relative Cost of 1mol Capacity with a Safety Buffer
Resin A	3.1216
Resin B	4.0469
Resin C	3.7674
Resin D	3.2799
Resin E	3.1105
Resin F	4.8222
Resin G	5.3600
Resin H	4.4233
Resin I	3.6958

## Data Availability

Restrictions apply to the availability of these data. Data was obtained based on commercial application testing and are available upon author contact with the permission of Sanhua-Aweco Appliance Polska sp. z.o.o. sp.k.

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
