# Peer review of "Evaluation of Strong Cation Ion-Exchange Resin Cost Efficiency in Manufacturing Applications—A Case Study"

_polymers, 2022, doi:10.3390/polym14122391_

Round 1

Reviewer 1 Report

In this manuscript, the effective ionic capacities of strong cation ion-exchange resins including 9 different products of equivalent TDS capacity were investigated and compared using conditions similar to those found in white goods, in order to establish behavioural differences between commercial products and evaluate their capacity in a broader business context. The standard deviation in the resins’ performance and their relative prices are also evaluated.

I consider the content of this manuscript will definitely meet the reading interests of the readers of the Polymers journal. Therefore, I suggest giving a minor revision and the authors need to clarify some issues or supply some more data to enrich the content.

  • For the Keywords, ‘effective ionic capacity’, ‘material applicability evaluation’, and ‘behavioural difference’ should also be added to attract a broader readership and highlight the significance of this work.

  • Please pay attention to grammar and spelling problems, especially the missing or redundant definite articles. I suggest double-checking the whole manuscript. I will point out several examples, but unfortunately, I cannot point out all of them. For example:

Line 35, ‘it was found that the use of softened water can allow for detergent savings of up to 70%’;

Line 152, ‘The purpose of the setup was to ensure a very high repeatability for the resin testing, which could also be realized by manual control methods, according to Fig.4’;

Line 199, ‘Each resin was subject to at least two runs, totalling at a minimum of 16 regeneration-softening cycles’;

Line 246, A comparison of the materials relative costs is shown in Table 2 and figure 7. Table 2. The relative cost of 1 kg for resin products under DAP Incotermsand so on.

  • Line 31, ‘as dirt removal has been observed to deteriorate in the presence of Ca2+ions [6].’ What is the mechanism of this phenomenon? Is it due to the formation of some side products? The reason should be explained briefly.

  • Line 42, ‘According to the data, membrane methods are the most popular for industrial applications [16]; however, the cost of these systems remains high in comparison to ion-exchange resins [17].’Here, the membrane methods should be clarified as the ‘nanofiltration membrane method’. And why is the nanofiltration membrane more expensive than ion-exchange resins?

    What is the chemical composition of the ion-exchange resins? This should also be introduced in detail. As far as I know, in energy storage applications, ion-exchange membranes are widely used, such as commercial Nafion membranes. They are perfluorosulfonic acid membranes, which are expensive, and even resin is very expensive. But they are cation exchange membranes with very high ion exchange capacity [Electrochimica Acta 378 (2021): 138133]. So the ion-exchange resins should be explained in detail and compared with the commonly known ion-exchange resins for energy applications, and it is a vital point to highlight the significance of this work.

  • Line 78, ‘All the suppliers’ names and the ion-exchange resin products’ brand names were omitted due to commercial considerations.It is understandable to omit the brand name, but at least nine samples belong to which kind of polymer structure, which should be briefly introduced or classified. For example, A-B-C belongs to perfluorinated sulfonic acid resin, C-D belongs to aromatic hydrocarbon resin, etc. Otherwise, what is the reference value of these nine resins to readers? Are these nine completely confused resins universally applicable?

  • Line 214, ‘The use of mass per submerged beaded resin volume data from the sample normalization procedure allowed us to recalculate the average molar operating capacity per litre of beaded resin to achieve the average molar operating capacity per kg of delivery-condition material.What is the difference in average molar operating capacity per kg compared to the ion-exchange capacity (meq/g)? Just one nominal (as provided by the company) and one during practical operation, or the situations are more complex?

Author Response

Dear Reviewer

Thank you for the thorough review.

Please see the attachment, containing a structured author's response to the reviews, divided into comment/suggestion column and corrections/answer column, per each review.

The change-tracking mode version of the manuscript will be uploaded as an attachment for the issue editor, as only one attachment can be uploaded to this reply.

Your sincerely

Maciej Kobielski

Reviewer 2 Report

The purpose of the paper is clearly stated to: “present a new method for strong acid cation-exchange resin comparison, which allows the raw material costs regarding production-level efficiency to be compared.”

I found it to be interesting and well-written. The resins are not identified “due to commercial considerations.”

“All the samples were strong acid cation-exchange resins of 1.9–2.0 eq capacity…” I assume the capacity is on a mass basis, rather than volume. Please specify. The moisture holding capacity is another value reported by the manufacturers; can you list that as well?

The “Ion-exchange testing cycle” is given in detail.

On page 7, please correct ETDA to EDTA

The method where “The per-kilogram price was divided by the per-kilogram delivery condition averaged molar capacity, to obtain the cost of the deionization efficiency for each material sample.” Is a useful quantitative measure.” Thus it follows that “The results of the molar capacity per kilogram are indicative of the best performing material from a chemical perspective. The industrial cost and engineering approach, however, requires the prices of the materials to be compared.”

Figure 7 (Comparison of relative cost of 1 kg for resin products…) is reasonable but Table 3 (Relative cost of 1 mmol capacity for resin products) shows numbers that are higher than I expected. No mention of Table 3 is made in the text. Please discuss how the numbers are calculated as you did with Figure 7.

Though the numbers you give are useful, you have the data to enhance the value of your report: Is there any difference among the resins in withstanding the regeneration cycle? Is there more or less fracturing among the different resin beads?

Author Response

(The authors gave the same response as above.)
